# Competing Endogenous RNA Networks in the Epithelial to Mesenchymal Transition in Diffuse-Type of Gastric Cancer

**DOI:** 10.3390/cancers12102741

**Published:** 2020-09-24

**Authors:** Natalia Landeros, Pablo M. Santoro, Gonzalo Carrasco-Avino, Alejandro H. Corvalan

**Affiliations:** 1Advanced Center for Chronic Diseases, Pontificia Universidad Católica de Chile, Santiago 8330034, Chile; natalialanderos@udec.cl (N.L.); pablosantoro@accdis.cl (P.M.S.); 2Advanced Center for Chronic Diseases, Universidad de Chile, Santiago 8380000, Chile; 3Department of Pathology, Hospital Clinico Universidad de Chile and Clinica Las Condes, Santiago 7550000, Chile; gcarrasa@me.com

**Keywords:** diffuse-type gastric cancer, epithelial to mesenchymal transition, E-cadherin, EMT-inducing transcription factors, microRNA, long non-coding RNA, competing endogenous RNA

## Abstract

**Simple Summary:**

The diffuse-type of gastric cancer is associated with epithelial to mesenchymal transition. Loss of E-cadherin expression is the hallmark of this process and is largely due to the upregulation of the transcription factors ZEB1/2, Snail, Slug, and Twist1/2. However, miRNA and lncRNAs can also participate through these transcription factors which directly target E-cadherin. The competing endogenous RNA (ceRNA) network hypothesis state that lncRNA can sponge the miRNA pool that targets these transcripts. Based on the lack of said networks in the epithelial to mesenchymal transition, we performed a prediction analysis that resulted in novel ceRNA networks which will expand our knowledge of the molecular basis of the diffuse-type of gastric cancer.

**Abstract:**

The diffuse-type of gastric cancer (DGC), molecularly associated with epithelial to mesenchymal transition (EMT), is increasing in incidence. Loss of E-cadherin expression is the hallmark of the EMT process and is largely due to the upregulation of the EMT-inducing transcription factors ZEB1/2, Snail, Slug, and Twist1/2. However, ncRNA, such as miRNA and lncRNAs, can also participate in the EMT process through the direct targeting of E-cadherin and other EMT-inducing transcription factors. Additionally, lncRNA can sponge the miRNA pool that targets these transcripts through competing endogenous RNA (ceRNA) networks. In this review, we focus on the role of ncRNA in the direct deregulation of E-cadherin, as well as EMT-inducing transcription factors. Based on the relevance of the ceRNA network hypothesis, and the lack of said networks in EMT, we performed a prediction analysis for all miRNAs and lncRNAs that target E-cadherin, as well as EMT-inducing transcription factors. This analysis resulted in novel predicted ceRNA networks for E-cadherin and EMT-inducing transcription factors (EMT-TFs), as well as the expansion of the molecular basis of the DGC.

## 1. Introduction

Gastric cancer (GC) is the fifth cause of incidence and the third-leading cause of cancer deaths worldwide [1]. Although variations in incidence and prevalence have been observed across geographical regions, high mortality rates are found in Asia, Eastern Europe, and particularly in Latin America [2,3]. GC is characterized by a combination of environmental factors, nutrition (diet), and infectious agents (i.e., *Helicobacter pylori* and Epstein-Barr virus) [4,5,6]. A major role is also played by the genetic architecture of the host and hereditary cancer susceptibility syndromes [7] (Figure 1).

Interactions of these factors result in two types of GC, the intestinal-type (IGC) and the diffuse-type (DGC). The former is preceded by clearly defined and sequentially ordered precancerous histological changes (i.e., atrophy and intestinal metaplasia) [8]. The latter, also defined as poorly cohesive (PC) carcinoma, including signet ring cell (SRC) histology [9,10], arises “de novo”. The existence of precancerous changes in the surrounding mucosa does not influence the prognosis of the DGC [11] (Figure 1).

The gross aspect of GC can be divided into four types according to the Borrmann classification: Polypoid (type I), fungating (type II), ulcerating (type III), and diffusely infiltrating carcinoma (type IV). Type I, II, and III represent around 60% of all GC, and greatly correspond to the IGC. Type IV is also referred to as ‘linitis plastica’, and usually corresponds to the DGC, according to Lauren classification [9] (Figure 2).

Histologically, as shown in Figure 2, the IGC is characterized by large glandular lumina accompanied by papillary fold formation and/or solid components, reminiscent of normal gastric or intestinal glands. On the other hand, the DGC is characterized by loosely (non-cohesive) arranged tumor cells with prominent mucin-filled intracytoplasmic droplets with an enlarged, eccentrically located, flattened nucleus (signet ring morphology). These cells infiltrate the lamina propria in between normal gastric glands without a desmoplastic reaction. In contrast to the IGC, an increase in incidence, a lack of effective systemic cytotoxic chemotherapy, and a strong heritage component are unique features of the DGC [12,13,14,15,16,17].

**Figure 2 cancers-12-02741-f002:**
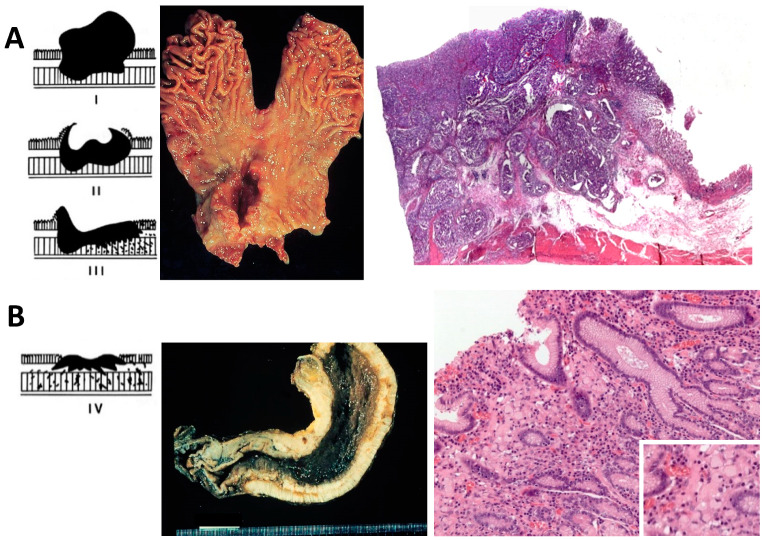
Schematic Borrmann classification, gross pathology, and histological view of intestinal- and diffuse-type gastric cancer. (**A**) Intestinal-type gastric cancer, the left panel shows schematic Borrmann types polypoid (type I), fungating (type II), and ulcerating (type III); the middle panel shows a total gastrectomy specimen with a Borrmann II gastric cancer in the antrum, lesser curvature, and anterior wall; right panel shows a histological scan view of gastric cancer forming glands and occasional papillary structures infiltrating up to the muscularis propria; non-tumor adjacent gastric mucosa is seen to the right. (**B**) Diffuse-type gastric cancer, the left panel shows schematic Borrmann classification with diffusely infiltrating carcinoma (type IV), the middle panel shows a total gastrectomy specimen in which the entire gastric wall is thickened, hard, and infiltrated by tumor; left panel shows a signet-ring cell diffuse-type gastric cancer with tumor cells with a mucin droplet pushing and flattening the nucleus to the side and infiltrating the lamina propria without a desmoplastic reaction; the inset shows a higher magnification (400×) of tumor cells. Schematic Borrmann classification was taken from Lecciones de Anatomía Patológica with permission [18].

The molecular basis of the DGC is beginning to be understood [19]. Deep molecular characterization of a large set of clinical samples has defined this sub-type of GC as genomically stable with low mutational burden, mostly restricted to the *CDH1* (E-cadherin) gene [20]. A strong correlation with an epithelial to mesenchymal transition (EMT) has been proposed by the Asian Cancer Research Group (ACRG), particularly in DGC tumors with signet ring cell features [21]. A recurrent loss of *CDH1* expression is also a common finding in this association. Of note, germline mutations in the *CDH1* gene have been identified in familial clusters of GC, particularly in hereditary diffuse gastric cancer (HDGC) syndrome [22].

## 2. The Molecular Pathology of the Epithelial-Mesenchymal Transition

EMT is an essential process during embryogenesis but is aberrantly reactivated in cancer in association with tumor progression and metastasis [23,24]. Differentiated gastric epithelial cells have epithelial cell-to-cell junctions and apical-basal polarity, a process highly associated with the expression of E-cadherin. In the EMT process, the epithelial cells lose the expression of E-cadherin and concomitantly upregulate mesenchymal-phenotype proteins such as N-cadherin, Vimentin, and Fibronectin [25]. Consequently, these cells feature increased growth and migratory capacities, acquire stem cell-like properties, and become highly invasive and metastatic [25] (Figure 3). Specific transcriptional repressors of *CDH1* expression have been identified. These repressors (ZEB1/2, Snail, Slug, and Twist1/2), also known as EMT-inducing transcription factors (EMT-TF), are frequently overexpressed in cancer and associated with poor survival [26,27,28]. When tumor cells reach metastatic sites, they perform the reverse process of EMT, known as a mesenchymal-epithelial transition (MET). This enables colonization through the upregulation of genes that encode epithelial-phenotype proteins [29].

### 2.1. E-CADHERIN

*CDH1* is a tumor suppressor gene that contains 16 exons (100 kb), located in the chromosome 16q22.1, and encodes for E-cadherin, a transmembrane glycoprotein of 120 kDa [30]. E-cadherin has three major structural domains: An extracellular domain forming the cell–cell adhesion (adherens junctions), a single transmembrane domain, and a cytoplasmic segment, that has binding domains for α, β-catenin, and p120 to form the cytoplasmic cell adhesion complex that interacts with the actin cytoskeleton [31]. E-cadherin is mainly found in the plasmatic membrane of epithelial cells, where play an important role in epithelial architecture, maintenance of polarity, and cellular differentiation [31]. E-cadherin is considered a sensor of the cell external environment. Additionally, the cadherin-catenin complex modulates various signaling pathways in epithelial cells such as Wnt/β-catenin, Rho GTPases, and NF-κB and participates in development and carcinogenesis [32]. The loss of E-cadherin expression is associated with poorly differentiated and poor prognosis tumors, as is the case of DGC [33,34]. In familial cancer syndromes (in particular HDGC [35]), the loss of *CDH1* function, due to germline mutations, confers an extremely high lifetime risk (>80%) of developing GC. The most common mutations are small insertions and deletions (35%) and large exon deletions, along with nonsense mutations (28%). Moreover, nonsense mutations (16%) and junction site mutations (16%) have been described [30].

### 2.2. ZEB

*ZEB* gene family (zinc finger E-box binding homeobox) comprises two major members, *ZEB1* and *ZEB2*, that are transcriptional repressors and epigenetically modulate the expression of *CDH1* in multiple human cancers [36]. ZEB1 represses *CDH1* transcription by binding to two E-box sequences from its promoter and recruiting DNMT1 (DNA methyltransferase) to maintain the methylated state of the promoter [37]. Moreover, it improves heterochromatinization in the promoters of multiple target genes. Particularly, ZEB1 interacts with the SWI/SNF chromatin-remodeling protein BRG1 to regulate E-cadherin [38]. Furthermore, ZEB1 represses transcription by recruiting CtBP1 (C-terminal-binding protein 1) as a co-repressor for numerous DNA-binding transcriptional repressors, including E-cadherin [39]. Dysregulation of ZEB1 and ZEB2 have been involved in tumor progression associated with the development of mesenchymal phenotype, stem-like properties, metastasis, and resistance to therapeutic agents [40].

### 2.3. SNAIL and SLUG

Snail (zinc finger transcription factor *SNAI1*) suppresses *CDH1* transcription by binding to 3 E-boxes of its promoter region [41] and playing a crucial role in the recruitment of repressor complexes on the E-cadherin promoter, such as DNMT, histone methyltransferases, and the histone demethylase LSD1 (lysine demethylase 1) [42]. In addition, Snail recruits histone deacetylase inhibitors and the co-repressor mSin3A, enriched in deacetylated histones H3 and H4, to the promoter region of *CDH1* [26]. The upregulation of Snail is inversely correlated with the expression of E-cadherin in clinical samples of DGC [43].

Slug is encoded by Snail zinc finger family 2 (*SNAI2*) and is found to be overexpressed in human cancers in association with poor prognosis in clinical studies [44]. Slug participates in EMT by the recruitment of LSD1 to the *CDH1* promoter and represses its transcription [45].

### 2.4. TWIST

The Twist family proteins are a basic helix-loop-helix transcription factor encoded by the *TWIST1* and *TWIST2* genes. High expression levels of these proteins decrease the expression of E-cadherin and induce cell motility through the upregulation of mesenchymal-phenotype proteins [46]. Twist recruits SET8, a histone methyltransferase with monomethylation activity of H4K20, decreasing the expression of E-cadherin [47]. In DGC, increased levels of Twist have been highly correlated with the upregulation of N-cadherin, but inversely associated with E-cadherin expression. Based on these findings, Twist1/2 was proposed as key players for the E- to N-cadherin switch in DGC [43].

## 3. Noncoding RNAs: Classification and Functions

Non-coding RNAs (ncRNAs) represent the largest portion of the human transcriptome [48,49]. There is a wide range of ncRNAs that play important roles in biological and pathological processes. Among these, microRNAs (miRNAs) and long non-coding RNAs (lncRNAs) are the most frequently associated with cancer processes.

miRNAs are a type of single-stranded, small non-coding RNA, 18-24 nucleotides in length [50]. The 5′end of the miRNA has 6 to 8 nucleotide fragments known as the “seed region”, which bind by base complementarity to the miRNA response elements (MREs) in the 3′UTR of the target mRNA. This binding induces endonucleolytic degradation and/or translation repression [51]. It is estimated that about 60% of the cells mRNAs have MRE, resulting in the expression of a large number of genes being modulated by these small RNAs [51,52].

LncRNAs are the largest class of ncRNAs, with transcripts greater than 200 nucleotides and functions that depend on their subcellular location [53]. LncRNAs can be found in both the nucleus and the cytoplasm and are classified into guides, dynamic scaffolds, and molecular decoys. LncRNAs participate in important regulatory roles in biological processes, such as X inactivation, imprinting, development, mRNA processing, epigenetic modifications, and organization of nuclear architecture [54,55].

In the cytoplasm, lncRNAs generate a large-scale of trans-regulatory crosstalk through their capacity to communicate with coding mRNA [55,56,57,58]. In this crosstalk and through the MREs, lncRNAs act as sponges to hijack miRNAs, resulting in the reduction of an available miRNA pool to target mRNAs [59]. This language was first hypothesized by Salmena et al. as competing endogenous RNA (ceRNA) networks [60]. In this hypothesis, miRNAs are the core of this communication, and lncRNA and mRNA can compete for the same pool of miRNAs [61,62]. In the EMT process, both types of ncRNAs participate through the direct targeting of E-cadherin and other EMT-TFs. Additionally, a few documented examples have shown that lncRNA can also sponge the miRNA pool that targets these transcripts through ceRNA networks [63].

### 3.1. ncRNA in the EMT Process

The relation between EMT and ncRNAs is well-established. In particular, miRNAs and lncRNAs have been described as promoting EMT through the direct targeting of genes associated with this pathway. The most critical EMT genes and their associated miRNAs and lncRNAs are described below. In addition, a few ceRNA networks associated with the EMT pathway have also been described (Figure 3).

### 3.2. miRNA, lncRNA, and ceRNA as Direct Regulators of E-Cadherin

The expression of *CDH1* is directly suppressed by miRNAs in GC. For example, the upregulation of miR-9-5p correlates with the downregulation of *CDH1* in clinical samples [64]. In vitro experiments have also shown that miR-9-5p promotes cell migration and invasion by directly targeting this gene [65]. miR-199a-5p also directly targets and downregulates *CDH1* expression. This miRNA is upregulated by the transcription factor SRF (serum response factor), modulating EMT in GC cells [66]. miR-217-5p is overexpressed in tumor tissues and directly targets *CDH1* [67]. Consequently, an increased proliferation and decrease of apoptosis have been observed in vitro. Of note, this miRNA has been found in exosomes from the plasma of patients providing a novel biomarker candidate for non-invasive assessment of GC [67]. miR-544a-5p directly targets the mRNA of *CDH1*, which leads to the downregulation of E-cadherin. Furthermore, the expression of miR-544a-5p induces the transcription of EMT-TFs, such as Snail and ZEB1 [68]. This miRNA reduces the protein level of AXIN2, a component of the WNT/β-catenin signaling pathway, allowing β-catenin to translocate to the nucleus and induce vimentin expression [68]. These findings suggest that miR-544a-5p induce EMT through two independent pathways, *CDH1* and WNT/β-catenin (Table 1).

**Table 1 cancers-12-02741-t001:** Summary of all miRNAs and mRNA interaction and their roles in the EMT process associated with diffuse-type gastric cancer.

miRNA	mRNA	Function	Ref.
**Upregulation miRNAs**			
miR-9	E-cadherin	Upregulation promotes EMT through the direct binding of E-cadherin	[64]
miR-199a	E-cadherin	Upregulation promotes EMT through the direct binding of E-cadherin	[66]
miR-217	E-cadherin	Upregulation promotes EMT through the direct binding of E-cadherin	[67]
miR-544a	E-cadherin	Upregulation promotes EMT through the direct binding of E-cadherin	[68]
**Downregulation miRNAs**			
miR-141	ZEB1/2	Downregulation promotes EMT by increasing ZEB1 and ZEB 2	[69]
miR-200c	ZEB1/2	Downregulation promotes EMT by increasing ZEB1 and ZEB 2	[69]
miR-200c	ZEB1/2	Downregulation promotes EMT by increasing ZEB1 and ZEB 2	[70]
miR-205-3p	ZEB1/2	Downregulation promotes EMT by increasing ZEB1 and ZEB2	[71]
miR-574-3p	ZEB1	Downregulation promotes EMT by increasingZEB1	[72]
miR-145	ZEB2	Downregulation promotes EMT by increasing ZEB2	[73]
miR-200b	ZEB2	Downregulation promotes EMT by increasing ZEB2	[74]
miR-338-3p	ZEB2	Downregulation promotes EMT by increasing ZEB2	[75]
miR-506	ZEB2	Downregulation promotes EMT by increasing ZEB2	[76]
miR-22	Snail	Downregulation promotes EMT by increasing Snail	[77]
miR-153	Snail	Downregulation promotes EMT by increasing Snail	[78]
miR-195	Snail	Downregulation promotes EMT by increasing Snail	[79]
miR-204	Snail	Downregulation promotes EMT by increasing Snail	[80]
miR-491	Snail	Downregulation promotes EMT by increasing Snail	[81]
miR-33a	Slug	Downregulation promotes EMT by increasing Slug	[82]
miR-124	Slug	Downregulation promotes EMT by increasing Slug	[83]
miR-203	Slug	Downregulation promotes EMT by increasing Slug	[84]
miR-506	Slug	Downregulation promotes EMT by increasing Slug	[85]
miR-15a-3p	Twist	Downregulation promotes EMT by increasing Twist	[86]
miR-16-1-3p	Twist	Downregulation promotes EMT by increasing Twist	[86]
miR-186	Twist	Downregulation promotes EMT by increasing Twist	[87]
miR-381	Twist	Downregulation promotes EMT by increasing Twist	[88]
miR-495	Twist	Downregulation promotes EMT by increasing Twist	[89]

LncRNA AGAP2-AS1 is upregulated in clinical samples and promotes tumorigenesis and progression in vitro [90]. This lncRNA is located in the nuclear compartment, suggesting that the mechanism that regulates E-cadherin takes place at the transcriptional level. AGAP2-AS1 can bind to RNA-binding proteins, such as LSD1 and EZH2 (enhancer of zeste Homolog 2). Of note, EZH2 is the functional enzymatic component of the Polycomb Repressive Complex 2, which is responsible for the epigenetic maintenance of development and differentiation [90]. Both proteins directly target the promoter region of *CDH1*. In this way, AGAP2-AS1 could participate as an epigenetic repressor of E-cadherin expression. As shown in Figure 3, several other lncRNAs (Lnc01614, LOC554202, TUG1, AFAP1-AS1, NEAT, MNX1-AS1, AOC4P, SNHG20, CCAT1, TPT1-AS1, and LINC00941) have been found to be upregulated and associated with poor prognosis in clinical samples of GC [91,92,93,94,95,96,97,98,99,100]. The knockdown of these lncRNAs was associated with the re-expression of E-cadherin. Since the cellular sublocalization has not been described, these lncRNAs may bind RNA binding proteins in the nucleus or activate EMT-TFs through ceRNA networks in the cytoplasm. The lncRNA MEG3 has been found downregulated in tumor tissues and cell lines. Transfection of this lncRNA upregulated E-cadherin expression and inhibited the expression of mesenchymal markers [101,102]. These findings suggest that the loss of lncRNA-MEG3 expression favors the progression of GC through the EMT process by as yet unknown mechanisms (Table 2).

To this date, only one ceRNA network involving E-cadherin has been reported. Chen et al. [103] identified that the lncRNA RP11-789C1.1 sponges miR-5003-3p, which directly binds *CDH1* transcripts by dual-luciferase reporter assays. Silencing of lncRNA RP11-789C1.1 reduced the expression of E-cadherin and promoted EMT in GC cells [103] (Table 3).

**Table 2 cancers-12-02741-t002:** Summary of all lncRNAs and mRNAs interactions and their roles in the EMT process associated with diffuse-type of gastric cancer.

lncRNA	mRNA	Function	Ref.
**Upregulated lncRNAs**			
AFAP1-AS1	E-cadherin	Upregulation of AFAP1-AS1 decreases E-cadherin through unknown mechanism and promotes EMT	[104]
AGAP2-AS1	E-cadherin	Upregulation of AGAP2-AS1 decreases E-cadherin through unknown mechanism and promotes EMT	[90]
AOC4P	E-cadherin	Upregulation of AOC4P decreases E-cadherin through unknown mechanism and promotes EMT	[98]
CCAT1	E-cadherin	Upregulation of CCAT1 decreases E-cadherin through unknown mechanism and promotes EMT	[92]
LINC00941	E-cadherin	Upregulation of LINC00941 decreases E-cadherin through unknown mechanism and promotes EMT	[99]
LNC01614	E-cadherin	Upregulation of LNC01614 decreases E-cadherin through unknown mechanism and promotes EMT	[97]
LOC554202	E-cadherin	Upregulation of LOC554202 decreases E-cadherin through unknown mechanism and promotes EMT	[105]
MNX1-AS1	E-cadherin	Upregulation of MNX1-AS1 decreases E-cadherin through unknown mechanism and promotes EMT	[94]
NEAT1	E-cadherin	Upregulation of NEAT1 decreases E-cadherin through unknown mechanism and promotes EMT	[95]
SNHG20	E-cadherin	Upregulation of SNHG20 decreases E-cadherin through unknown mechanism and promotes EMT	[93]
TPT1-AS1	E-cadherin	Upregulation of TPT1-AS1 decreases E-cadherin through unknown mechanism and promotes EMT	[91]
DNM3OS	Snail	Upregulation of DNM3OS promotes EMT through unknown mechanism and increases Snail	[106]
HOTAIR	Snail	Upregulation of HOTAIR promotes EMT through unknown mechanism and increases Snail	[107]
MALAT1	Snail	Upregulation of MALAT1 promotes EMT through unknown mechanism and increases Snail	[108]
XLOC_010235	Snail	Upregulation of XLOC_010235 promotes EMT through unknown mechanism and increases Snail	[109]
LINC00978	Slug	Upregulation of LINC00978 promotes EMT through unknown mechanism and increases Slug	[110]
Linc-GPR65-1	Slug	Upregulation of Linc-GPR65-1 promotes EMT through unknown mechanism and increases Slug	[111]
FRLnc1	Twist	Upregulation of FRLnc1 promotes EMT through unknown mechanism and increases Twist	[112]
ZFAS1	Twist	Upregulation of ZFAS1 promotes EMT through unknown mechanism and increases Twist	[113]
TUG1	EZH2	Upregulation of TUG1 promotes development of GC by increasing EZH2	[100]
**downregulated lncRNAs**			
MEG3	E-cadherin	Downregulation of MEG3 decrease E-cadherin through unknown mechanism and promotes EMT	[101]
LINC00261	Slug	Downregulation of LINC00261 prevents the degradation of Slug favoring EMT	[114]

**Table 3 cancers-12-02741-t003:** Summary of all ceRNAs interactions and their roles in the EMT process associated with diffuse-type gastric cancer.

lncRNA	State	miRNA	State	mRNA	State	Function	Ref.
RP11-789C1.1	Down	miR-5003-3p	Up	E-cadherin	Down	RP11-789C1.1/miR-5003-3p/E-cadherin axis promotes EMT	[103]
CASC15	Up	miR-33a	Down	ZEB1	Up	CASC15/miR-33a/ZEB1 axis promotes EMT	[115]
CAT104	Up	miR-381	Down	ZEB1	Up	CAT104/miR-381/ZEB1 axis promotes EMT	[116]
H19	Up	miR-141	Down	ZEB1	Up	H19/miR-141/ZEB1 axis promotes EMT	[117]
MAGI2-AS3	Up	miR-141-3p	Down	ZEB1	Up	MAGI2-AS3/miR-141-3p/ZEB1 axis promotes EMT	[118]
MAGI2-AS3	Up	miR-200a-3p	Down	ZEB1	Up	MAGI2-AS3/200a-3p/ZEB1 axis promotes EMT	[118]
SNHG6	Up	miR-101-3p	Down	ZEB1	Up	SNHG6/miR-101-3p/ZEB1 axis promotes EMT	[119]
ZEB1-AS1	Up	miR-149-3p	Down	ZEB1	Up	ZEB1-AS1/miR-149-3p/ZEB1 axis promotes EMT	[120]
UCA1	Up	miR-203	Down	ZEB2	Up	UCA1/miR-203/ZEB2 axis promotes EMT	[121]
GCMA	Up	miR-34a	Down	Snail	Up	GCMA/miR-34a/SNAIL axis promotes EMT	[122]
GCMA	Up	miR-124	Down	Snail	Up	GCMA/miR-124/SNAIL axis promotes EMT	[122]
H19	Up	miR-22-3p	Down	Snail	Up	H19/miR-22-3p/Snail1 axis promotes EMT	[123]
PVT1	Up	miR-30a	Down	Snail	Up	PVT1/miR-30a/SNAIL axis promotes EMT	[124]
SNHG7	Up	miR-34a	Down	Snail	Up	SNHG7/miR-34a/SNAIL axis promotes EMT	[125]
XIST	Up	miR-101	Down	EZH2	Up	XIST/miR-101/EZH2 axis promotes EMT	[126]
lncRNA-ATB	Up	miR-141-3p	Down	TGF-β2	Up	lnc-ATB/miR-141-3p/TGF-β2 axis promotes EMT	[127]

### 3.3. miRNA, lncRNA, and ceRNA in the Regulation of the EMT-Inducing Transcription Factors

#### 3.3.1. ZEB

The expression of ZEB1 and ZEB2 has been reported to be controlled by miRNAs and lncRNAs. One of the most studied miRNA families in EMT is miR-200 (miR-200a, miR-200b, miR-200c, miR-141, miR-429), which targets the ZEB1 and ZEB2 repressors [128]. Downregulation of miR-200b is associated with aggressiveness variables, such as tumor size, depth of invasion, lymphatic vessel invasion, and lymph node metastasis. Overexpression of miR-200b reverses proliferation, invasion, and migration in GC cells through direct targeting of ZEB2 [74]. miR-200b contributes to chemoresistance in DGC, particularly to vincristine (VCR), cisplatin (CDDP), etoposide (VP-16), adriamycin (ADR), and 5-Fluoracil (5-Fu) [129]. Downregulation of miR-200c is associated with the upregulation of TGF-β (transforming growth factor-β), a major inducer of EMT [70]. Furthermore, miR-200c and miR-141 regulate EMT through the direct binding of ZEB1 and ZEB2 [69]. Other miRNAs (miR-145-5p, miR-205-3p, miR-338-3p, miR-506, and miR-574) have been found downregulated in clinical samples and are associated with a poor survival rate. In GC cells, these miRNAs reverse migration and invasion and participate in EMT and invasion by directly targeting both ZEB1 and ZEB2 [71,73,75,76]. The underlying mechanisms of the loss-of-expression of these miRNAs warrant further investigation. miR-145 is also a contributor to chemoresistance in DGC, particularly to 5-Fu [129] (Table 1).

At least nine lncRNAs associated with ceRNA networks have been reported for ZEB. CASC15 expression is associated with proliferation, migration, EMT, and tumorigenesis in vivo. In clinical samples, its overexpression is associated with a poor prognosis. This lncRNA upregulates ZEB1 expression by sponging miRNA-33a-5p [115]. Another important lncRNA is H19, which negatively correlates with miR-141-5p expression in GC cells and tissues. It has been shown that H19 sponges miR-141-5p, upregulating ZEB1 expression, and favoring the EMT process [117]. The lncRNA-ATB is overexpressed in GC tumors and associated with lymph node metastasis [127]. This lncRNA acts as a ceRNA by sequestering miR-141-5p, which, in addition to increasing the levels of ZEB1, also upregulates TGF-β, favoring the EMT and cancer progression [127,130]. MAGI2-AS3 was identified as an EMT-related lncRNA and is highly co-expressed with ZEB1 and ZEB2 in both tumor and normal stomach tissues. The upregulation of MAGI2-AS3 predicts a poor prognosis. This lncRNA sponges miR-200a-3p and miR-141-3p, which enables the upregulation of ZEB1 and ZEB2 [118]. UCA1 is one of the most frequently upregulated lncRNAs in GC. This upregulation is associated with lymph node metastasis and higher TNM stage [121]. UCA1 represses the expression of *CDH1* through ZEB2 expression through the sponging of miR-203 [121]. CAT104 is also highly expressed in tissues and promotes cell viability, migration, and invasiveness in vitro. This lncRNA sequesters miR-381, which directly targets ZEB1 [116]. SNHG6 is overexpressed in clinical samples and has a pro-metastatic function by sponging miR-101-3p and increasing ZEB1 expression [119]. Accordingly, the expression of miR-101-3p is lower in tumor tissues than in normal gastric mucosa. This downregulation is associated with tumor progression and promotes cell invasion and migration in vitro [131]. LncRNA XIST regulates ZEB expression by acting as a molecular sponge for miR-101-5p [126]. The upregulation of XIST in clinical samples is associated with a higher TNM stage and poor survival rate [126]. The lncRNA ZEB1 antisense 1 (ZEB1-AS1) is transcribed in conjunction with the *ZEB1* gene by sharing a bidirectional promoter. ZEB1-AS1 sponges miR-149-3p and enable the upregulation of ZEB1 and ZEB2, promoting migration and invasiveness in GC cells [120] (Table 3).

#### 3.3.2. SNAIL

miR-153 has been reported as a tumor suppressor gene downregulated in GC. This downregulation is inversely correlated with Snail expression. Bioinformatic analysis and luciferase reporter assay showed that this miRNA targets the 3′UTR of Snail transcript [78]. miR-204 suppresses metastasis and invasion in GC cells, regulating EMT by directly binding the Snail1 mRNA [80]. miR-204 contributes to chemoresistance in DGC, particularly to 5-Fu and oxaplatin (L-OHP) [129]. miR-491-5p mediates Snail suppression and expression levels and is inversely correlated in tumor tissues. Downregulation of miR-22-5p and miR-491-5p have been found in clinical samples. The silencing of these miRNAs induces EMT and promotes metastasis in vitro [77,81]. As is the case with ZEB, the mechanisms underlying the loss-of-expression of these four miRNAs warrants further investigation. miR-195-5p also targets Snail and is downregulated in GC cells. Interestingly, propofol (2, 6-diisopropylphenol), an anesthetic agent widely used in the clinic, can inhibit the survival and growth in vitro [132,133]. Functional assays demonstrate that miR-195-5p is upregulated in the presence of propofol, suppressing EMT, invasion, and migration of GC cells [133]. This observation deserves further study, including animal trials and prospective clinical studies [134] (Table 1).

In clinical samples, overexpression DNM3OS correlates to worse overall survival. This lncRNA promotes tumor progression and upregulate Snail, and promotes EMT in vitro [106]. DNM3OS is localized in the nucleus, and its expression is induced by Twist [135,136]. MALAT1 (Metastasis-Associated Lung Adenocarcinoma Transcript 1) is upregulated in clinical samples of GC and associated with depth of invasion, higher TNM stage, and prognostic marker for metastatic disease [137]. As nuclear-localized, an epigenetic transcriptional function is the most plausible molecular mechanisms of inducing EMT of MALAT1 [138]. Another lncRNA that regulates Snail is HOTAIR (HOX Antisense Intergenic RNA). This lncRNA is transcribed from the antisense strand of the *HoxC* gene and localized in the nucleus, as well as in the cytoplasm of the cell [139]. Overexpression of HOTAIR is associated with lymph node metastasis and poor 5-year survival rate [107]. HOTAIR promoted EMT through the upregulating of Snail [107] by yet unknown mechanisms. XLOC_010235, a 302 bp lncRNA, is overexpressed and associated with tumor proliferation, invasion, and metastasis in GC cells. This lncRNA promotes EMT by down-regulating E-cadherin and the concomitantly upregulation of mesenchymal-phenotype proteins XLOC_010235 positively regulated Snail at mRNA and protein levels [109]. The subcellular localization of XLOC_010235 has not yet been reported (Table 2).

Four studies have reported ceRNA networks for Snail. H19 expression in tumor tissues has been inversely correlated with miR-22-3p and associated with cell growth and metastasis. These effects were enhanced by the upregulation of Snail [123]. Consequently, a ceRNA network has been proposed for H19/miR-22-3p/Snail axis [123]. The lncRNA PVT1 has been found upregulated in clinical samples. This lncRNA promotes migration and invasion in vitro, and tumor growth and metastases in vivo. LncRNA PVT1 sponges miR-30a and prevents degradation of Snail. Consequently, the axis PVT1/miR-30a/Snail is another example of ceRNA network [124]. GCMA (Gastric Cancer metastasis-associated lncRNA) is overexpressed in GC. GCMA enhances the migratory capacities, invasiveness, and metastases in vitro and in vivo, respectively. GCMA acts as a ceRNA through the sponging of miR34a that targets Snail. Consequently, this lncRNA promotes EMT and invasiveness in GC [122]. LncRNA SNHG7 is also upregulated in tumor tissues and represses miR-34a by direct binding. In vitro, this lncRNA enhanced migration and invasiveness. The inverse correlation of the expression levels of miR-34a and Snail suggests that SNHG7 upregulate EMT by miR-34a/Snail axis [125] (Figure 3). miR-34a contributes to chemoresistance in DGC, particularly to CDDP, doxorubicin (DOX), docetaxel (DTX), and gemcitabine (GEM) [129] (Table 3).

#### 3.3.3. SLUG

Loss-of-expression of miR-33a increased mRNA and protein expression of Slug and mesenchymal-phenotype proteins, while decreasing the expression of E-cadherin in GC tissues [82]. The expression of miR-203 has been reported to be inversely correlated with the expression of Slug in tumor tissues, and directly bind this transcript in vitro [84]. Downregulation miR-506 is associated with poor overall 5-year survival in cases of GC. Luciferase reporter assays have shown that this miRNA binds to the 3′UTR of Slug and increased expression of E-cadherin, suppressing cell proliferation, and migration [85] (Table 1).

At least three lncRNAs have been described in association with Slug. Linc-GPR65-1, is localized in the nucleus and regulates at the transcriptional level the expression of Slug. This lncRNA has been found upregulated in primary tumors and promotes proliferation, migration, and cell invasion in vitro. Linc-GPR65-1 indirectly regulates Slug through the PTEN-AKT pathway favoring invasion and migration [111]. LINC00978, also known as MIR4435-2HG or AK001796, has been associated with the EMT process. In clinical samples, LINC00978 is associated with a higher TNM stage and lymph node metastasis. In vitro, this lncRNA is localized in the cell nucleus and promotes tumor progression by regulating the EZH2-mediated silencing of E-cadherin expression [110]. Another lncRNA that regulates Slug is Linc00261, which is downregulated in clinical samples and negatively correlated with a poor 5-year survival rate. Linc00261 is located in the cytoplasmatic compartment and inhibited the EMT process and metastasis in vitro and in vivo, respectively, by the upregulating E-cadherin. Of note, GSK3-β phosphorylates Slug to be ubiquitinated and degrades by the proteasome [140], and consequently, Linc00261 improves the interaction between GSK3-β and Slug by promoting its degradation [114] (Table 2).

Only one ceRNA has been reported for Slug, GCMA, that acts as a ceRNA through the sponging miR-124 that directly binds Slug [83]. lncRNA GCMA is found to be overexpressed in clinical samples and enhances the migratory and invasiveness of GC cells in vitro and in vivo, respectively [122] (Table 3).

#### 3.3.4. TWIST

Twist expression is also regulated by ncRNAs, such as miR-15a-3p and miR-16-1-3p. Both miRNAs are associated with the abnormal regulation of Twist1 and the EMT process in GC development [86]. Of note, miR-16-1-3p contributes to chemoresistance in GC, particularly to VCR, CDDP, VP-16, ADR, 5-Fu, and mitomycin (MMC) [129]. miR-495 inhibits proliferation and metastasis and promotes apoptosis by targeting Twist1 in GC cells [89]. miR-186 affects the proliferation, invasion, and migration of GC by inhibiting Twist1 [87]. The upregulation of miR-381 causes a decrease in the expression of Twist1 at the mRNA and protein levels in GC cells [88] (Table 1).

A few studies have shown lncRNAs associated with Twist. The lncRNA ZFAS1 (Zinc finger antisense 1) has been localized in the nucleus and in the cytoplasm [141]. Clinically, it has been found upregulated in tumor tissues, as well as peripheral blood samples from patients with GC. This upregulation positively correlates with the upregulation of Twist, as well as EMT-TF [113]. ZFAS1 can also be found in exosomes from GC patients. Of note, when GC cells were treated with exosomes loaded with ZFAS1, an improvement in proliferation and migration along with the upregulation of Twist was observed [142]. This finding suggests that ZFAS1 could promote EMT, although its mechanisms are unknown. FRLnc1 is another lncRNA associated with the upregulation of Twist and cell migration. In addition, this lncRNA also induce the expression of the master regulator of EMT, TGFβ1 [112]. As mention before, HOTAIR promoted EMT not only through the upregulation of Snail, but also inducing the expression of Twist in GC cells [107] (Table 2). Since the subcellular localization of this lncRNA is dual, nucleus, and cytoplasm, both epigenetic and ceRNA mechanisms should be analyzed. In the case of Twist, ceRNA networks are yet to be defined.

## 4. Prediction of ceRNAs for EMT Processes

As the largest portion of the human transcriptome, lncRNAs play emerging roles in cancer processes, such as EMT [48,49]. In addition to their important regulatory roles in the nucleus, lncRNAs participate in the cytoplasm through the ceRNA networks along with miRNAs and coding mRNAs. miRNAs comprise the core of these ceRNA networks. On one side of the core, miRNAs target and promote mRNA decay, whereas on the other side, they are sponged by lncRNAs and are no longer available for binding mRNA. Based on the relevance of the ceRNA network hypothesis and the lack of said networks in EMT, we performed a prediction analysis for all miRNAs and lncRNAs that target E-cadherin, as well as EMT-TF. The miRwalk 2.0 database was utilized to predict binding sites for miRNAs:lncRNAs and miRNAs:mRNAs [143,144].

This database was searched for complementary seed sequences between miRNA and the entire sequence of the lncRNA and miRNA and the 3′UTR region of mRNA in at least 3 out of 5 and 4 out of 12 algorithms, respectively. This search resulted in 29 novel miRNAs and 34 novel lncRNAs associated with EMT that may build novel ceRNA networks for E-cadherin, as well as EMT-TFs. As shown in Figure 4 and Appendix A, this analysis resulted in 147 ceRNAs networks, 18 for E-cadherin, 50 for ZEB1/2, 41 for Snail, 9 for Slug, and 29 for Twist1/2. Given that no effective systemic cytotoxic chemotherapy has, thus far, been established, it is worth noting that 62 of these networks contained six previously described chemoresistant-associated miRNAs. Pending experimental validation, these results may translate into novel therapeutic approaches in DGC. Ultimately, this analysis did not predict any ceRNA networks for eight lncRNAs (linc-GPR65-1, LINC00978, XLOC_010235, lnc01614, CAT104, AOCA4P, FRlnc1, and LOC554202), which are validated regulators of E-cadherin or EMT-TFs. These findings indicate that additional bioinformatics tools may be necessary for a comprehensive discovery of the ceRNA networks that regulate the EMT process.

## 5. Conclusions

Stemming from the remarkable increase in DGC incidence, the emerging role of the EMT process as the molecular basis of DGC has become an active area of research. In the EMT process, gastric cells lose the expression of E-cadherin, due to the upregulation of the EMT-TFs ZEB1/2, Snail, Slug, and Twist1/2. As the largest portion of the human transcriptome, miRNAs directly suppress E-cadherin expression and lncRNAs sponge, and hijack the miRNAs which directly target the EMT-TFs. This phenomenon represents the ceRNA network hypothesis. Based on the relevance of this hypothesis and the scarceness of these networks in EMT, we conducted a prediction analysis that resulted in 147 novel ceRNAs networks worthy of further investigation. The discovery of several ceRNA networks containing chemoresistant-associated miRNAs opens a translational window into potential innovations in systemic therapies in DGC. The lack of predictions in the case of a few lncRNAs warrants investigation with additional bioinformatic tools. This review is a comprehensive overview of the scarce number of ceRNA networks in the EMT process, as well as a proposal for novel ceRNA networks in the DGC.

## Figures and Tables

**Figure 1 cancers-12-02741-f001:**
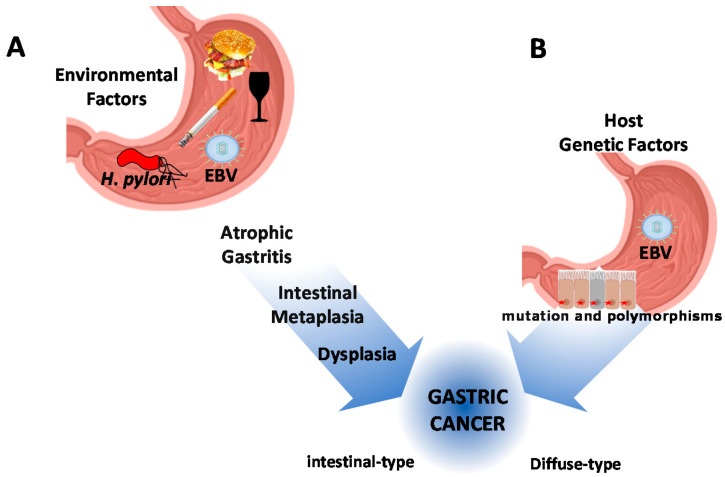
Schematic representation of gastric cancer pathogenesis. (**A**) The contribution of environmental factors, nutrition, and infectious agents. Chronic infection by *Helicobacter pylori* is considered the main infectious agent in combination with diet, smoking, and alcohol. All these conditions drive the precancerous cascade of the intestinal-type of gastric cancer. (**B**) Host genetic factors, including hereditary cancer susceptibility syndromes, drive the carcinogenesis process of diffuse-type gastric cancer. The Epstein-Barr virus (EBV) latent infection has been described as associated with both subtypes of gastric cancer (GC).

**Figure 3 cancers-12-02741-f003:**
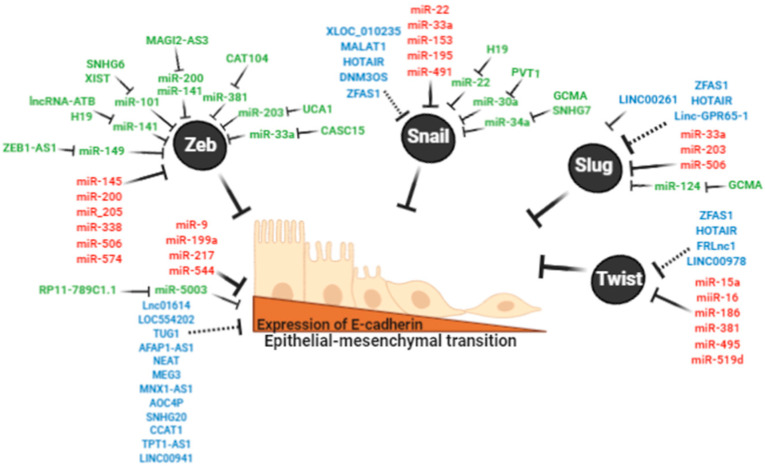
Non-coding RNA regulates the expression of E-cadherin and epithelial to mesenchymal transition (EMT)-inducing transcription factors (EMT-TFs). Black circles show EMT-TFs ZEB1/2, Snail, Slug, Twist that directly regulate E-cadherin. In red, miRNAs that directly bind *CDH1* and EMT-TFs transcripts. In blue, lncRNA that directly bind *CDH1* and EMT-TFs (AGAP2-AS1, HOTAIR, LINC00261) or dysregulate these transcripts through potential ceRNAs. In green, reported ceRNA networks to *CDH1* and EMT-TFs transcripts.

**Figure 4 cancers-12-02741-f004:**
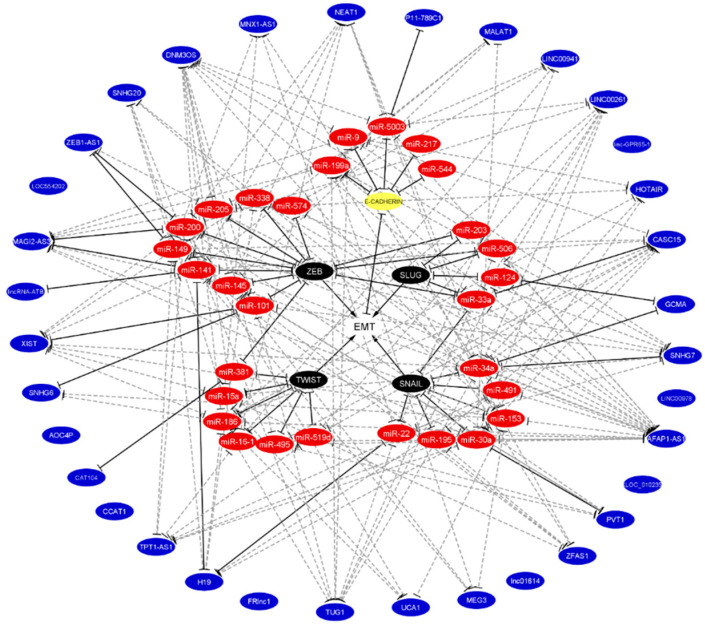
Integration of predicted and validated non-coding RNA that regulates EMT the binding with E-cadherin and EMT-TFs. Validated miRNA, lncRNA, and ceRNA networks were taken from literature according to Figure 3. Predicted miRNA, lncRNA, and ceRNA networks were predicted by the miRwalk 2.0 database. The white circle denotes EMT, the yellow circle denotes E-cadherin, black circles denote EMT-TFs, red circles denote miRNAs, and blue circles denote lncRNAs. Validated bindings are shown in solid black lines, while predicted bindings are shown in gray dotted lines. The lack of lines in blue circles denotes that no predictions were found. Of note, the chemoresistant miRNAs (miR-15a-3p, miR-16-1-3p, miR-34a, miR-141, miR-145, and miR-200) participate in these networks.

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
