# Peer review of "Competing Endogenous RNA Networks in the Epithelial to Mesenchymal Transition in Diffuse-Type of Gastric Cancer"

_cancers, 2020, doi:10.3390/cancers12102741_

Round 1

Reviewer 1 Report

The review by Landeros et al. is interesting, well organized and presented.

I have only few concerns about it:

  • The different paragraphs need to be numbered.
  • Line 109. The sentence ‘In familial cancer syndromes, in particular, HDGC [29], loss of CDH1 function, due to germline mutations confers an extremely high lifetime risk (>80%) of developing GC’ needs to be rephrased as follows ‘In familial cancer syndromes, in particular HDGC [29], loss of CDH1 function, due to germline mutations, confers an extremely high lifetime risk (>80%) of developing GC’.
  • Line 147. The paragraph is mostly aimed to present Non coding RNAs functions rather than their proper action in EMT. I suggest to change the title in ‘Non coding RNAs: classification and functions’.
  • Referring to the previous point, a scheme or figure, explaining the molecular mechanisms through which ceRNAs regulate gene expression, would ameliorate the paper and benefit the reader in its comprehension.
  • At my opinion, Conclusion section appears weaker with respect to the rest of the manuscript. For instance, Authors could better discuss the possible future applications, and concurrent limitations, on the use of ceRNAs as potential candidate for GC treatment and therapy.

Author Response

Reviewer 1. The different paragraphs need to be numbered.

Line 109. The sentence ‘In familial cancer syndromes, in particular, HDGC [29], loss of CDH1 function, due to germline mutations confers an extremely high lifetime risk (>80%) of developing GC’ needs to be rephrased as follows ‘In familial cancer syndromes in particular HDGC [29], loss of CDH1 function, due to germline mutations, confers an extremely high lifetime risk (>80%) of developing GC’. 

Answer, thanks for this comment. We have change the mentioned paragraph for the suggested sentence: "In familial cancer syndromes in particular HDGC [34], loss of CDH1 function, due to germline mutations, confers an extremely high lifetime risk (>80%) of developing GC."

Line 147. The paragraph is mostly aimed to present Noncoding RNAs functions rather than their proper action in EMT. I suggest changing the title in ‘Noncoding RNAs: classification and functions’. 

Answer, thanks for this comment. We have changed the tittle to: “Noncoding RNAs: classification and functions” as suggested.

Referring to the previous point, a scheme or figure, explaining the molecular mechanisms through which ceRNAs regulate gene expression, would ameliorate the paper and benefit the reader in its comprehension.

Answer, we appreciate this observations. According to guidelines of the journal, we have added a graphical abstract focusing in the regulation of EMT through ceRNA networks. 

In my opinion, the Conclusion section appears weaker with respect to the rest of the manuscript. For instance, the Authors could better discuss the possible future applications, and concurrent limitations, on the use of ceRNAs as a potential candidate for GC treatment and therapy.
Answer. thanks for this suggestion. We have added a paragraph about future applications. This paragraph state "The discovery of several ceRNA networks containing chemoresistant-associated miRNAs opens a translational window into potential innovations in systemic therapies in DGC". 

Reviewer 2 Report

Overall recommendation:

 Minor change

Final comments:

 This paper shows detailed analysis of how competing endogenous RNA networks regulate EMT in diffuse-type of gastric cancer (DGC). Because DGC is a highly chemoresistant cancer, no effective systemic cytotoxic chemotherapy has been established. Surgical resection is effective only at an early stage.

 Please discuss how these knowledge contribute new type of therapy. Such discussion will attract clinicans.

Author Response

Reviewer 2. This paper shows a detailed analysis of how competing endogenous RNA networks regulate EMT in diffuse-type of gastric cancer (DGC).

Because DGC is a highly chemoresistant cancer, no effective systemic cytotoxic chemotherapy has been established. Surgical resection is effective only at an early stage. Please discuss how this knowledge contributes to a new type of therapy. Such discussion will attract clinicians.

Answer, thanks for this comment. We have added the following paragraph to answer this comment: "In contrast to the IGC, an increase in incidence, a lack of effective systemic cytotoxic chemotherapy, and a strong heritage component are unique features of the DGC [12-17]". 

Reviewer 3 Report

Landeros et al. present a manuscript reviewing and collating data on non-coding RNAs involved in the EMT and diffuse-type gastric cancer in particular. At the end of the paper they also perform some data mining for new potential lncRNAs with a role in above mentioned processes.

The manuscript is well written and organised. I particularly like the introduction to the gastric cancer field. I do have few comments that should be considered:

  1. A table summarising all miRNAs and lncRNAs, together with their roles and levels in gastric cancer is needed. As it stands all information is scattered across the manuscript with Figure 3 providing only a crude overview of the networks.
  2. The relation between EMT and miRNAs have been extensively studied. This should be reflected when the EMT is first introduced. 
  3. Importantly, MET should be also introduced as it plays a significant role in cancer dissemination.
  4. Figure 1 looks cut at the bottom.

Author Response

Reviewer 3. The manuscript is well written and organized. I particularly like the introduction to the gastric cancer field. I do have a few comments that should be considered:

A table summarising all miRNAs and lncRNAs, together with their roles and levels in gastric cancer is needed. As it stands all information is scattered across the manuscript with Figure 3 providing only a crude overview of the networks. 

Answer, thanks for this comment: We have added a summary table (Table 1) with all interactions of lncRNAs, miRs and mRNA and their roles in the EMT process.

The relation between EMT and miRNAs have been extensively studied. This should be reflected when the EMT is first introduced. 

Answer, we appreciated this suggestion. We have introduced a paragraph to refelct the relation between EMT and ncRNAs as follow:

"The relation between EMT and ncRNAs is well-established. In particular, miRNAs and lncRNAs have been described as promoting EMT through the direct targeting of genes (by directly targeting genes) associated with this pathway (Fig. 3). The most critical EMT genes and their associated miRNAs and lncRNAs are described below. In addition, a few ceRNA networks associated with EMT pathway have also been described."

Importantly, MET should be also introduced as it plays a significant role in cancer dissemination. 

Answer, Thanks for this comment. We have added the next paragraph about MET in cancer dissemination: 

When tumor cells reach metastatic sites, they perform the reverse process of EMT known as a mesenchymal – epithelial transition (MET). This enables colonization through the upregulation of genes that encode epithelial [28]

Figure 1 looks cut at the bottom. 
Answer, thanks for this observation. We correct this mistake.